# Suppression of Ribose-5-Phosphate Isomerase a Induces ROS to Activate Autophagy, Apoptosis, and Cellular Senescence in Lung Cancer

**DOI:** 10.3390/ijms23147883

**Published:** 2022-07-17

**Authors:** Yu-Chin Nieh, Yu-Ting Chou, Yu-Ting Chou, Chao-Yung Wang, Shi-Xian Lin, Shih-Ci Ciou, Chiou-Hwa Yuh, Horng-Dar Wang

**Affiliations:** 1Institute of Biotechnology, National Tsing Hua University, Hsinchu 300044, Taiwan; judy0872@gmail.com (Y.-C.N.); ytchou@life.nthu.edu.tw (Y.-T.C.); lss339840@gmail.com (S.-X.L.); asb3851@gmail.com (S.-C.C.); 2Division of Hematology and Oncology, Department of Medicine, Helen Diller Comprehensive Cancer Center, University of California, San Francisco, CA 94158, USA; yu-ting.chou@ucsf.edu; 3Department of Cardiology, Chang Gung Memory Hospital, Linkou Main Branch, Chang Gung University, Taoyuan 33305, Taiwan; cwang@ocean.ag; 4Institute of Molecular and Genomic Medicine, National Health Research Institutes, Zhunan 35053, Taiwan; 5Institute of Bioinformatics and Structural Biology, National Tsing Hua University, Hsinchu 300044, Taiwan; 6Department of Biological Science and Technology, National Yang Ming Chiao Tung University, Hsinchu 300093, Taiwan; 7Ph.D. Program in Environmental and Occupational Medicine, Kaohsiung Medical University, Kaohsiung 807378, Taiwan; 8Institute of Systems Neuroscience, National Tsing Hua University, Hsinchu 300044, Taiwan; 9Department of Life Science, National Tsing Hua University, Hsinchu 300044, Taiwan

**Keywords:** ribose-5-phosphate isomerase A, autophagy, apoptosis, cellular senescence, lung cancer

## Abstract

Ribose-5-phosphate isomerase A (RPIA) regulates tumorigenesis in liver and colorectal cancer. However, the role of RPIA in lung cancer remains obscure. Here we report that the suppression of RPIA diminishes cellular proliferation and activates autophagy, apoptosis, and cellular senescence in lung cancer cells. First, we detected that RPIA protein was increased in the human lung cancer versus adjust normal tissue via tissue array. Next, the knockdown of RPIA in lung cancer cells displayed autophagic vacuoles, enhanced acridine orange staining, GFP-LC3 punctae, accumulated autophagosomes, and showed elevated levels of LC3-II and reduced levels of p62, together suggesting that the suppression of RPIA stimulates autophagy in lung cancer cells. In addition, decreased RPIA expression induced apoptosis by increasing levels of Bax, cleaved PARP and caspase-3 and apoptotic cells. Moreover, RPIA knockdown triggered cellular senescence and increased p53 and p21 levels in lung cancer cells. Importantly, RPIA knockdown elevated reactive oxygen species (ROS) levels. Treatment of ROS scavenger N-acetyl-L-cysteine (NAC) reverts the activation of autophagy, apoptosis and cellular senescence by RPIA knockdown in lung cancer cells. In conclusion, RPIA knockdown induces ROS levels to activate autophagy, apoptosis, and cellular senescence in lung cancer cells. Our study sheds new light on RPIA suppression in lung cancer therapy.

## 1. Introduction

Lung cancer is one of the leading causes of cancer death worldwide [1]. Non-small cell lung cancer (NSCLC) is the most common type of lung cancer, which can be further divided into three major types: squamous cell carcinoma (SCC), adenocarcinoma, and large cell carcinomas. The risk factors of lung cancer include smoking, environmental, hormonal, and genetic factors [2]. Approximately 10~40% lung cancer patients do not have any history of tobacco smoking [3]; the population of lifelong non-smokers of lung cancer patients is much larger in Asia than in Western countries [4]. Furthermore, women are more susceptible to developing lung cancer than men. Therefore, understanding the pathophysiology of lung cancer and developing the therapeutic means are urgent.

Aging is an irreversible physiological process. During the progression, aging also promotes hyperplastic pathologies, leading to cancer [5]. Genomic instability is associated with cancer and aging [6]. Several aging-related genes, such as mTOR signaling [7] and p53 [8], also play critical roles in cancer. In addition, aging causes unstable genomes and DNA damage, and abnormal proliferation promotes tumor formation [9]. Aging and cancer are also connected to metabolic dysregulation. The pentose phosphate pathway (PPP) is a major glucose metabolic pathway which meets the cellular demands of biosynthesis and antioxidant defense. It has been suggested that the activation of PPP enhances cancer cell growth [10]. PPP is composed of oxidative and non-oxidative phases for the production of NADPH and ribose-5-phosphate. PPP has several functions in protecting against oxidative stress and in hypoxia [11].

Ribose-5-phoshate isomerase A (RPIA), the key regulator of the non-oxidative branch of PPP, involves nucleotide metabolism and RNA, DNA synthesis. RPIA can convert ribulose-5-phosphate (Ru5P) to ribose-5-phosphate (R5P) [12]. Decreased nucleotide metabolism leads to cell senescence in the early stage of tumor formation [13]. Kras promotes glycolysis intermediates and non-oxidative PPP in pancreatic tumors, and blocking the non-oxidative PPP by the knockdown of *RPIA* or *RPE* in the tumor cells inhibits tumor growth [14]. In addition, RPIA is involved in liver cancer [15,16], colorectal cancer [17,18] and endometrial cancer [19]. However, the role of RPIA in lung cancer remains unclear.

Autophagy is a catabolic process that can mediate the lysosomal degradation of defective organelles, long-lived proteins, and a variety of protein aggregates. Autophagy begins with the formation of the phagophore, in which proteins of the cytosol and complete organelles are engulfed by double-membrane structures known as autophagosomes or early autophagic vacuoles. Upon fusion with the lysosome, autophagosomes form single-membrane structures called autolysosomes or late autophagic vacuoles, and the resultant elements are returned into cytosol for metabolic reactions [20]. Increased LC3-II and decreased p62/SQSTM1(p62) are two markers of autophagy. The autophagosome membrane is built by a kind of LC3-phosphatidylethanolamine, LC3-II. Hence, LC3-II accumulation is used as an autophagy marker. On the other hand, p62 is degraded by autophagy and may serve to link ubiquitinated proteins to the autophagic machinery [21,22,23].

Autophagy maintains cell survival under different stress conditions. Additionally, autophagy is a form of non-apoptotic (type II) programmed cell death. In the animal model, the knockout of p62 results in decreased tumorigenesis [24]. The overexpression of autophagy-related genes, *atg5*, inhibits tumor growth and extends the lifespan in mice [25], suggesting a possible role of autophagy in tumorigenesis. RPIA regulates autophagy by inhibiting LC3 processing [26]; however, whether lung cancer is associated with RPIA mediated autophagy remains vague.

Cellular senescence is an irreversible cell arrest and is characterized by a flattened and enlarged cellular morphology. Many stimuli conduct cellular senescence including telomere shortening, DNA damage and oxidative stress [27]. There are two major effector pathways leading to cellular senescence: the p53 pathway and pRB pathway. Upon mitogenic stress or DNA damage, the tumor suppressor protein p53 is activated to induce the expression of p21, the cyclin-dependent kinase inhibitor, and consequently trigger cell cycle arrest [28]. In the second pathway, the retinoblastoma protein (pRB) is activated by p16 upon DNA damage or cellular stress, which inhibits cell cycle progression, leading to senescence [29]. Besides cellular morphology, there are several methods for the detection of senescence, such as increased senescence-associated β-galactosidase (SA-β-gal) activity and elevated p53 and p21 expression. Senescence limits the cell ability to proliferate and reduces lifespan. The tumor-suppressive function of cellular senescence was reported previously; TP53 mutation switching oncogene-induced senescence from suppressor to driver in primary tumorigenesis was revealed in a zebrafish mosaic analysis [30]. Nonetheless, cellular senescence is considered an important target for cancer treatment [31].

Over the past decades, the activation of apoptosis has evolved into an efficient approach for inhibiting cancer and has become a major target of cancer therapy. Apoptosis or programmed cell death (PCD) is a fundamental cell death which occurs when DNA damage is beyond repair. Apoptosis can happen by extrinsic and intrinsic pathways: the extrinsic pathway is initiated by death-receptor stimulation, and the intrinsic pathway is characterized by mitochondrial outer membrane permeabilization (MMP) and the release of mitochondrial cytochrome c [32]. The abnormal expression of some apoptosis key regulatory factors may lead to cancer, indicating the complicated relationships between apoptosis and cancer [33]. Those intracellular apoptosis signals produced in response to cellular stresses, such as DNA damage and oxidative damage caused by reactive oxygen species (ROS), and extrinsic inducers of apoptosis include growth factor hormones and toxins. The activation of caspases 3 plays an important role in both pathways and is used as an apoptosis indicator.

ROS are chemically reactive oxygen-containing molecules, include oxygen anions, free radicals, such as superoxide (O_2_^−^), hydroxyl radical (OH^−^), and H_2_O_2_ [34]. ROS is involved in many forms of cell deaths [35]. For instance, autophagy activation resulted in an increase in ROS [36]. On the other hand, excessive ROS induced apoptosis through both the extrinsic and intrinsic pathways. In intrinsic pathways, ROS promotes cytochrome c release by activating pore-stabilizing proteins (Bcl-2 and Bcl-xL). ROS activates JNK by increasing the p-JNK levels and in turn, induces extrinsic and intrinsic apoptotic signaling [37,38,39]. Thus, ROS plays an important role to activate programmed cell death pathways, including apoptosis, autophagy and cellular senescence.

Our laboratory previously identified a longevity mutant fly line with reduced expression of *ribose-5-phosphate isomerase (rpi)*. The knockdown of *rpi* in neurons extends the lifespan and attenuates polyglutamine toxicity-induced neurodegeneration in *Drosophila* [40]. During the anti-aging effect to anti-cancer, we found that human *ribose-5-phosphate isomerase A* (*RPIA*) was up-regulated in the tumors of hepatocellular carcinoma (HCC) patients. *RPIA* overexpression regulates cell proliferation and colony formation ability, and modulates tumor growth in nude mice [15] and the overexpression of RPIA in hepatocyte-induced liver cancer formation in zebrafish [16]. We also identified that the non-canonical function of RPIA activates β-catenin in colorectal cancer [18] and activates ERK and β-catenin pathways in hepatocarcinogenesis [16]. However, the role of RPIA in lung cancer remains concealed. In this study, we demonstrated that the knockdown of *RPIA* increases ROS levels to activate autophagy, apoptosis, and cellular senescence in lung cancer cells. Our study sheds new light on RPIA suppression in lung cancer therapy.

## 2. Results

### 2.1. Increased RPIA Expression Is Detected in the Tumor Biopsies of Lung Adenocarcinoma Patients

Our previous studies showed that RPIA levels were up-regulated in the tumor biopsies and the tumor tissue arrays of HCC patients and colorectal cancer patients. To explore whether RPIA also participates in the lung cancer, we used immunohistochemistry staining analysis to examine the RPIA expression levels using the lung adenocarcinoma tumor paired tissue array (Figure 1). Elevated RPIA expression levels were detected in the tumor tissues (Figure 1a–c) compared to the adjacent normal tissues (Figure 1d–f) in the lung adenocarcinoma paired tissues. The detailed complete data also supported the phenomena as shown in Appendix A and the detail in Appendix A. The results indicate that RPIA expression is significantly increased in the lung adenocarcinoma biopsies, suggesting that RPIA also plays a role in lung tumorigenesis.

### 2.2. Knockdown of RPIA Decreases Colony Formation Ability and Cell Proliferation in Lung Cancer Cells

Given that RPIA was up-regulated in the tumor biopsy of lung adenocarcinoma, subsequently, we investigated whether the knockdown of RPIA reduces tumor cell proliferation and inhibits colony formation by using the A549 cell line, which is lung carcinoma epithelial cells. We used the lentivirus infection system to establish two independent RPIA-shRNA knockdown cell lines (named shRPIA#1 and shRPIA#2) in A549 cells. The mRNA levels of RPIA in shRPIA#1 and shRPIA#2 A549 cells were decreased about 80% compared to that in the scrambled (Sc) control A549 cells by quantitative real-time PCR (Figure 2a). The RPIA protein levels were significantly decreased about 65% to 78% in both shRPIA knockdown cells lines compared to the control (Figure 2b). The results demonstrated that the two established lentiviral RPIA knockdown A549 cell lines display a significant reduction in RPIA expression.

Successively, we analyzed whether the suppression of RPIA can reduce colony formation ability and cell proliferation in lung cancer cells. The knockdown of RPIA significantly reduced about 50% of the colony number in both shRPIA#1 and shRPIA#2 cells compared to that in the Sc control cells (Figure 3a,b). Furthermore, we inspected the effect of RPIA knockdown on cell proliferation in shRPIA#1 and shRPIA#2 A549 cells on days 1, 2, 4, and 6 by WST-1 assay. Our results showed that the knockdown of RPIA in shRPIA#1 and shRPIA#2 A549 cells significantly diminished the cell proliferation ability, compared to the control cells (Figure 3c).

Activated MAPK/ERK (mitogen-activated protein kinase/extracellular signal-regulated kinase) signaling promotes cell proliferation, cell survival and metastasis in tumors [41,42,43]. Our previous studies indicated that the knockdown of RPIA can reduce cell proliferation and colony formation ability in liver cancer cells by modulating ERK signaling [15]. Therefore, we examined whether the knockdown of RPIA can also reduce the levels of phosphorylated ERK1/2 (p-ERK1/2) in shRPIA#1 and shRPIA#2 A549 lung cancer cells. The results demonstrated that the knockdown of RPIA decreased p-ERK1/2 protein levels in both shRPIA knockdown cells (Figure 3d). We also investigated whether the overexpression of RPIA increases the levels of p-ERK1/2 in A549 lung cancer cells. Overexpression of RPIA indeed increased p-ERK1/2 protein levels, and the treatment of MEK inhibitor PD98059 abolished the increased levels of p-ERK 1/2 by RPIA overexpression (Figure 3e). These data imply that RPIA can increase the levels of p-ERK1/2, and enhance the colony formation ability and cell proliferation in A549 lung cancer cells. We also observed that the levels of p-ERK1/2 were reduced by the knockdown of RPIA in two other lung cancer cell lines: H23 lung cancer cells (p53^M246I^ and KRAS^G12C^) and H358 lung cancer cells (p53^-null^ and KRAS^G12C^) (Appendix A). Since A549 harbors wild-type p53 and KRAS^G12S^, RPIA regulates p-ERK1/2 levels regardless of the genetic background.

### 2.3. Knockdown of RPIA Induces Apoptosis in Lung Cancer Cells

Our previous studies showed that the reduced liver cancer cell proliferation by RPIA knockdown is associated with enhanced apoptosis [15]. Hence, we questioned whether the knockdown of RPIA also induces apoptosis in A549 cells. We measured apoptosis cells using flow cytometry by Annexin V/PI double staining. The results showed that the knockdown of RPIA increased the percentages of apoptotic cells detected by Annexin V/PI double staining (Figure 4a). The quantification of Annexin V-positive and PI-positive apoptotic cells displayed 2–3 fold increments by RPIA knockdown compared to the Sc control cells (Figure 4b). Western blot analysis also indicated that the levels of pro-apoptotic markers, such as phosphorylated JNK (p-JNK), Bax, PUMA, c-PARP (cleaved form) and caspase 3 (active form), were elevated by RPIA knockdown; in contrast, the levels of Bcl-2, anti-apoptotic protein, were compromised (Figure 4c). Together, these data demonstrate that the suppression of RPIA facilitates mitochondria-mediated apoptosis in lung cancer cells.

### 2.4. Suppression of RPIA Promotes Autophagy in Lung Cancer Cells

Upon RPIA knockdown in A549 cells, we observed some vacuole morphology similar to autophagic phenotype. Thus, we investigated whether the knockdown of RPIA triggers autophagy in lung cancer cells. We found that the knockdown of RPIA formed more vacuoles than the control (Figure 5a). By using acridine orange staining, we revealed that the knockdown of RPIA induced more acidic vacuole formation in lung cancer cells (Figure 5b). Furthermore, enhanced GFP-LC3 punctae were observed in the stable expression of GFP-LC3 in shRPIA#1 and shRPIA#2 (Figure 5c), indicating that RPIA knockdown activates autophagy. The statistical analysis indicated that the percentages of LC3-GFP punctae positive cells detected in shRPIA#1 and shRPIA#2 cells were 36% and 84%, respectively, which are much higher compared to the Sc control with 26% (Figure 5d). Furthermore, the suppression of RPIA increased the levels of LC3-II and reduced the expression of p62 in both shRPIA#1 and shRPIA#2 cells by Western blot (Figure 5e), further supporting that the knockdown of RPIA induces autophagy in A549 lung cancer cells. We also observed p62 levels decreased upon RPIA knockdown in H23 and H358 lung cancer cell lines (Appendix A). Hence, RPIA activates autophagy in lung cancer cells in different genetic backgrounds. To determine the role of RPIA knockdown-induced autophagy, we applied autophagy inhibitors (3-MA and CQ) in the *RPIA* knockdown cells; both treatments resulted in more cells to apoptotic cell death in lung cancer (Appendix A). Thus, we deduced that the activated autophagy by RPIA knockdown may serve as cytoprotection from cell death.

To confirm that RPIA knockdown induces autophagy in A549 lung cancer cells, we utilized transmission electron microscopy (TEM) to examine the cellular ultrastructure of the A549 lung cancer cells with and without RPIA knockdown. The results revealed that autophagic isolation membrane, autophagosome (Ap), and lysosome (Ly) were presented in RPIA-silenced A549 but not in the control (Sc) cells (Figure 6). Together, these data further support that the knockdown of RPIA induces autophagy in A549 lung cancer cells.

### 2.5. Knockdown of RPIA Triggers Cellular Senescence and Elevates p53 and p21 Expression Levels in A549 Lung Cancer Cells

In addition to apoptosis and autophagy induced by *RPIA* knockdown, we also observed that the knockdown of *RPIA* in A549 lung cancer cells resulted in a broad and flattened cellular phenotype similar to senescence (Figure 7a), and exhibited increased senescent cells with enhanced senescence-associated β-galactosidase (SA β-gal) activity (Figure 7b). It has been shown that p21 promotes cell cycle arrest in response to many stimuli, and regulates both pRB and p53 senescence pathways [44]. Thus, we examined whether the knockdown of *RPIA* induces p21 expression, and found that *RPIA* silencing up-regulates p21 mRNA and protein expression levels in A549 lung cancer cells (Figure 7c,d), also in H23 and H358 lung cancer cells (Appendix A). These data support that the knockdown of *RPIA* induces cellular senescence in lung cancer cells, regardless of their genetic background. Moreover, we found that the knockdown of *RPIA* significantly enhances p53 protein expression in A549 cells (Figure 7e,f), supporting the notion of the activation of p53/p21 axis in controlling cellular senescence by RPIA knockdown.

### 2.6. Reactive Oxygen Species Induced by RPIA Knockdown Result in Autophagy, Apoptosis and Cellular Senescence in A549 Cells

Previous studies indicated that excessive reactive oxygen species (ROS) induce autophagy and mitochondria apoptosis formation [45,46]. Accordingly, we inspected whether knockdown of *RPIA* induces apoptosis and autophagy by increasing ROS levels. Using flow cytometry with dichlorofluorescein diacetate (DCFH-DA) staining to measure ROS levels in the Sc control, shRPIA#1, and shRPIA#2 cells, we discovered higher ROS levels in the *RPIA* knockdown A549 cells than those in the control cells (Figure 8a). In addition, the treatment of *N*-acetyl-cysteine (NAC), a ROS scavenger, decreased the elevated ROS levels in the shRPIA knockdown A549 cells (Figure 8a). Next, we interrogated whether NAC treatment can revert the levels of the apoptosis and autophagy molecular markers mediated by the knockdown of *RPIA* in the A549 cells. The treatment of NAC seized the increased p21 levels for cellular senescence, the upraised levels of cleaved PARP (c-PARP) for apoptosis, the up-regulated LC3-II and down-regulated p62 levels for autophagy by the suppression of *RPIA* in A549 cells (Figure 8b). These data indicate that the knockdown of *RPIA* induces autophagy, apoptosis and cellular senescence mediated by rising ROS levels in lung cancer cells.

## 3. Discussion

Previously, our lab proclaimed that RPIA is up-regulated in liver cancer and colorectal cancer biopsies and regulates tumorigenesis [15,18], and activates β-catenin in colorectal cancer and enhances ERK and β-catenin in hepatocarcinogenesis [16,18]. However, the effect of RPIA in lung cancer is undisclosed. This study provided evidence that RPIA expression is also up-regulated and associated with tumor progression in lung adenocarcinoma cancer biopsies. In addition, we uncovered the molecular mechanisms by which the suppression of RPIA can increase ROS levels to induce apoptosis, autophagy, and cellular senescence for decreased cell proliferation and lowered colony formation ability in lung cancer cells (Figure 9).

Altered metabolism is a hallmark of cancer, and recently it has been vigorously investigated in cancers. Glucose metabolism and PPP pathway promote cancer formation by providing the energy and bio-materials required for tumor rapid growth [47]. In addition to glycolysis, alternative glucose metabolic pathways may induce a malignant phenotype [48]. For example, the activation of PI3K/AKT leads to glucose uptake, glycolysis and tumorigenesis in cancer cells [49]. Increasing studies portrayed the close relationship between altered metabolism and tumorigenesis. RPIA is a key regulator in the non-oxidative branch of the pentose phosphate pathway, and is involved in nucleotide biosynthesis, making RNA and DNA required for uncontrolled tumor growth. Previous studies also discovered that the altered PPP pathway activities by the dysregulated expression of transketolase like 1 (TKTL-1) or transaldolase (TALDO) is involved in tumor growth [50,51]. TKTL-1 was found to be up-regulated and correlated with cell progression and metastasis formation in colon cancer [52]. Higher TALDO expression was detected in the squamous cell carcinoma tumors of the head and neck cancer and in the lung epithelial carcinoma of smokers than non-smokers [53]. Therefore, we focused on the studies between the RPIA metabolism pathway and lung cancer. Consistent with the notion, our results showed that RPIA was up-regulated in the tumor biopsy of lung cancer patients, providing another line of important evidence that PPP plays an important role in the regulation of tumor growth upon dysregulation.

Accordingly, our results demonstrate that the knockdown of *RPIA* could inhibit cell growth by mediating mitochondria apoptosis. Recently, some chemotherapeutic drugs induced DNA damage, such as cisplatin, by causing the cross-linking of DNA, which ultimately triggers apoptosis [54]. We hypothesized that the knockdown of *RPIA* may lead to some kind of DNA damage to enhance ROS levels to induce apoptosis, autophagy, and cellular senescence to interfere cell proliferation and colony formation in lung cancer cells.

Autophagy can maintain macromolecular synthesis and ATP production. It is activated in response to physiological and pathological stimuli, including hypoxia, nutrient deprivation, growth factor depletion, oxidative damage, and metabolic stress [55]. Autophagy can be regarded as an important survival mechanism in normal and cancer cells. It has been shown that autophagy-deficient mice are more likely to develop tumors. Previous studies showed that the overexpression of Atg5 activates autophagy and extends the lifespan in mice [25]. In addition, beclin 1 is a tumor-suppressor gene that can cause the induction of autophagy and inhibition of tumorigenesis [56,57]. Interestingly, we found the knockdown of *RPIA* triggered autophagic vacuoles similar to autophagy cell morphology and decreased the clonogenic formation and cell proliferation rate in A549 lung cancer, suggesting that *RPIA* could participate in autophagy process and promote tumorigenesis. The study in HeLa cells by Heintze et al. also supports our findings [26].

Autophagy plays a role of self-eating or self-killing in cancer is unknown. Under certain condition, autophagy constitutes a stress adaptation that avoids cell death; whereas in other cellular situation, it constitutes an alternative cell-death pathway [58]. In this study, we found that knockdown of *RPIA* induces autophagy and apoptosis in A549 lung cancer. When we inhibit autophagy by autophagy inhibitors (3-MA and CQ) in the *RPIA* knockdown cells, it leads more cells to apoptotic cell death in lung cancer (Appendix A). Therefore, the *RPIA* knockdown-activated autophagy may serve as protection from cell death. Some studies revealed that inhibitors of autophagy could increase the anti-tumor effects of PFKFB3 inhibitors [59]. Other studies indicated that the inhibition of autophagy enhanced the efficiency of chemotherapeutic drugs [20].

A recent paper also mentioned the relationship between autophagy and anti-cancer therapy. Some autophagy inhibitors, such as the antimalarial drug chloroquine (CQ), was investigated in clinical trials and enhanced the efficiency of chemotherapeutic [60]. Another study reveled that chloroquine (CQ) can induce apoptosis in glioma in a xenograft model [61]. VP-BEZ235, another autophagy inhibitor, is a novel and potent dual PI3K/mTOR inhibitor which results in anti-proliferative and anti-tumor activity in cancer cells [62]; it is currently in clinical trials for advanced solid tumors [63]. There are many studies to clarify that the autophagy inhibitor plays an important role in cancers. Therefore, we may develop a RPIA inhibitor as anti-cancer therapeutics, combined with autophagy inhibitor or chemotherapy. Perhaps it can be used as an effective anti-cancer strategy.

In summary, we demonstrated that the knockdown of *RPIA* decreased the cell proliferation rate and colony formation ability via the p-ERK pathway and induced ROS to activate apoptosis and autophagy and promote cellular senescence in A549 lung cancer cells. Our results may shed light on alternative cancer therapy, according to the impaired cell metabolism.

## 4. Materials and Methods

### 4.1. DNA Plasmid Generation

For lentivirus-mediated knockdown of *RPIA*, the two short hairpin RNA (shRNA) expression constructs were purchased from the National RNAi Core Facility at Academia Sinica. The target sequence in p-LKO.1-shRPIA#1 (No. TRCN0000049410) is 5′-GAATTGGAAGTGGTTCTACAA-3′, and in p-LKO.1-shRPIA#2 (No. TRCN0000049409) is 5′-GCTGATGAAGTAGATGCTGAT-3′. For lentivirus-mediated overexpression of *RPIA*, the full-length *RPIA* cDNA was subcloned into *Sal*I and *Eco*RI sites of HR-puro vector as described previously [64,65], and the resultant construct was named HR-puro-RPIA.

### 4.2. Cell Culture

Human lung cancer cell lines A549 and H23 and H358 were maintained in RPMI-1640 supplemented with 10% fetal bovine serum (FBS), 100 units/mL of penicillin, 100 μg/mL of streptomycin, and 2 mM of L-glutamine solution, in a 37 °C, 5% CO_2_ incubator.

### 4.3. Immunohistochemistry

The lung adenocarcinoma tissue array (Biomax, LC1504) was used for immunohistochemistry. The slides were deparaffinized in xylene for 10 min, rehydrated in an ethanol gradient (100%, 90%, 80%), and heated at 95 °C for 10 min in 10 mM sodium citrate buffer (pH 6.0) in an autoclave for antigen retrieval. The slides were washed with PBS and incubated in 3% H_2_O_2_ for 20 min at room temperature. The sections were blocked in 1.5% normal horse serum and incubated at 4 °C overnight with an anti-RPIA antibody (1:100 dilution, Abnova, cat# H00022934-B01P). Finally, the slides were stained with a biotin-conjugated horse anti-mouse secondary antibody. Antigens were visualized using the ABC reagent kit (by VECTASTAIN Elite ABC kit) and DAB detection (by Invitrogen liquid DAB substrate kit). Then the slides were stained with H&E for histological characterization of lesion. The slides were counterstained with hematoxylin, dehydrated, mounted, and scanned with multiple focus layers in the Taiwan Mouse Clinic and analyzed by Panoramic Viewer software.

### 4.4. Lentiviral Infection

Lentiviral particles were produced by the co-transfection of pLKO.1-shRPIA#1 or pLKO.1-shRPIA#2 (for *RPIA* knockdown) or pLKO.1 (as shScramble control) separately with pCMV-delta 8.9 and pVSVG plasmids into HEK293T cells (2 × 10^6^ cells in a 10 cm dish). Transfection reagents that include 2.5 M calcium chloride (CaCl_2_) and HEBS (HEPES, NaCl, Dextrose, KCl, and Na_2_HPO_4_) were pre-mixed and vortexed for 15 s before adding into the cells. The virus-containing supernatant was collected after 48 h of transfection. Lentiviral infection for A549 cells was performed by adding virus solution and 8 μg/mL polyprene for 12 h. After infection, the virus-containing medium was removed, and the RPMI containing 2 μg/mL puromycin was added for at least 48 h. The puromycin-selected A549 cells had shRPIA knockdown or the scramble-shRNA control. For the preparation of the lentivirus-infected A549 stable cells with *RPIA* overexpression and the control cells, the similar methods as described above were used with HR-puro-RPIA (for *RPIA* overexpression) and HR-puro (as control).

### 4.5. RNA Extraction and Quantitative Real-Time PCR

RNA was extracted from cultured cells using TRIzol (Invitrogen) according to the manufacturer’s protocol. In order to isolate RNA from the whole cell lysate, BCP (Sigma Aldrich) was used as the delaminated solvent. In order to completely lysate the cells, the cells were incubated at room temperature for 5 min after adding 1 mL TRIzol. Cell lysates were transferred into eppendorfs. The 1 ml lysate was mixed with 100 μL BCP and agitated by vortex. After the above steps, the lysate was placed at room temperature for 5 min. Then, each eppendorf was centrifuged at 14,000 rpm at 4 °C for 15 min. After centrifugation, the aqueous of each eppendorf was transferred into a new eppendorf. In order to precipitate the total RNA, the same volume of isopropanol (Sigma Aldrich) was added into each tube and incubated at room temperature for 30 min. After 30 min incubation, each RNA sample was centrifuged at 14,000 rpm at 4 °C for 15 min. The supernatants were drained from each tube. Each sample was washed by 1 mL 75% alcohol/DEPC solution and centrifuged at 12,000 rpm at 4 °C for 10 min. The RNA pellets were occurred after above steps. To dissolve the total RNA, the proper volume of DEPC water was added. RNA quantification was performed by UV Spectrometer (Nano-drop 100). An equal amount of the extracted total RNA was then reverse transcribed into cDNA using the oligo-(dT)_12–18_ primer with SuperscriptIII reverse transcriptase (Invetrogen).

Ten-fold dilutions of each cDNA were prepared for subsequent PCR amplification with the Universal Probe Library system using Lightcycler480 instrument (Roche Applied Science, Mannheim, Germany). Primer sequences were designed to detect specific genes as listed in Table 1.

### 4.6. Protein Extraction and Western Blot

To extract total protein, all cells were seeded onto 60 mm culture dishes after 24 h culture. All cells were washed twice with PBS. Cells were lysed in whole cell extract (WCE) lysis buffer containing 20 mM HEPES 4-(2-hydroxyethyl)-1-piperazineethanesulfonic acid) (pH 7.5), 75 mM NaCl, 2.5 mM MgCl_2_, 0.1 mM EDTA, 0.5% Triton X-100, 0.1 mM Na3VO4, 50 mM NaF and protease inhibitor. Cell lysates were vibrated for 30 min and centrifuged at 13,200 rpm for 20 min at 4 °C. The supernatant was transferred to a new tube, and then analyzed for the concentration of each sample; the 5X protein dye was diluted into 1X as the final concentration and mixed with each sample, and the protein concentration was measured.

For Western blotting, all samples were diluted into equal protein concentration through adding the appropriate volume of RIPA buffer. Mixtures were heated at 95 °C for 5 min and loaded at 40 μg per lane. The protein was separated in a 12.5% SDS-PAGE gel and transferred to a nitrocellulose membrane. After blocking with 5% skim milk in TBST (tris-buffered saline containing 0.1% Tween 20), the membrane was incubated with a specific primary antibody (as listed in Table 2) overnight at 4 °C. After washing three times for 10 min, the membrane was incubated with HRP-conjugated secondary antibody for 1 h at room temperature (goat anti-mouse IgG 1:5000 and goat anti-rabbit IgG 1:5000) and washed three times for 10 min. To visualize the results, enhanced chemical luminescence (ECL) (Millipore, cat #WBKLS0500) was prepared by mixing, incubated with the membrane for 1 min, and the membrane was exposed to an X-ray film. Antibodies used in Western blot were listed in Table 2.

### 4.7. Colony Formation Assay

For colony formation with A549 lung cancer cells, 5 × 10^2^ cells/well were seeded in a 6-well cell culture dish. After 10 day of incubation (37 °C and 5% CO_2_), each well was washed by PBS gently. And each well was stained with 0.25% crystal violet diluted by methanol at room temperature for 30 min. After staining, each well was washed several times extensively until the background became limpid, and the total colony number was counted.

### 4.8. Water-Soluble Tetrazolium Salt-1 (WST-1)

A total of 1 × 10^3^ cells per well were seeded in 96-well plates. At the indicated time point, the culture medium was replaced with a WST-1 containing medium and incubated for 1 h. The absorbance at 490 nm was measured using a microplate reader (BIO-RAD model 680).

### 4.9. Transmission Electron Microscopy (TEM)

The cells were cultured at a 10 cm dish at 37 °C and the cells blocks were immersed in 2.5% glutaraldehyde/PBS for 1 h in the dark. The cells were washed three times for 10 min with PBS. The samples were incubated in 1% osmium tetroxide in PBS for 2 h to post-fixation in the dark. Then the samples were washed six times for 20 min in H_2_O and overnight. The samples were block stained in 0.1% uranyl acetate/H_2_O for 1 h in the dark. The cells were washed three times for 10 min in H_2_O. Next, the samples underwent block staining, dehydration, and resin infiltration. All samples were examined with JEOL JEM 1200EX.

### 4.10. Acridine Orange Staining

*RPIA* knockdown as well as the control knockdown A549 cells were stained with acridine orange at ten-thousand dilution. The cells were incubated at 37 °C for 15 min. After staining, the cells were visualized and photographed under an inverted microscope LX71 OLYMPUS.

### 4.11. Apoptosis Cells Detection

The analysis of apoptosis cells was performed by FACS (BD) in IBMS at Academia Sinica. The detection reagent was performed by the FITC Annexin V Apoptosis Detection kit II (BD Pharmingen^TM^). For the cells prepared for analysis, the cells were first washed twice with cold PBS and then resuspended in 1X Binding Buffer at a concentration of 1 × 10^6^ cells/mL, then 100 μL of the resuspended cell solution was transferred to an eppendorf, and 5 μL of FITC Annexin V and 5 μL PI were added to each tube and the cells were gently vortexed and incubated for 30 min at room temperature in the dark. Afterward, 400 μL of 1× Binding Buffer was added to each tube. The cells were further analyzed by flow cytometry within 1 h.

### 4.12. Senescence Assay

To measure senescence-associated-β-galactosidase activity, cells were incubated in a staining solution (5 mM citric acid, 10 mM sodium-phosphate, 150 mM sodium chloride, 2 mM magnesium chloride, 5 mM potassium ferrocyanide, 5 mM potassium ferricyanide, 1 mg/mL X-Gal, pH6.0) for 24 h at 37 °C. The cells were washed and embedded in PBS, viewed in an inverted transmission microscope (Leitz, Labovert FS) and documented by a digital imaging system (Nikon-Coolpix 990).

### 4.13. Reactive Oxygen Species (ROS) Cells Detection

The analysis of ROS cells was performed by FACS (BD). The detection reagent was performed by ENZO (Catalog No: ENZ-51011). For the cells prepared for analysis, the cells were first washed twice with 1X ROS buffer, then resuspended in 1X ROS buffer at a concentration of 2.5 × 10^5^ cells/mL, and transferred to an eppendorf and centrifuged at 4000 rpm for 5 min. Then the supernatant was discarded and DCFH-DA (1:5000 dilution) was added; then, it was incubated in the dark for 30 min at room temperature. The cells were further analyzed by flow cytometry within 1 h.

### 4.14. Statistical Analysis

The two-tailed Student *t* test was used for statistical analysis. All data were presented as the mean ± standard error of the mean (s.e.m.). A *p*-value less than 0.05 was considered to be significant.

## Figures and Tables

**Figure 1 ijms-23-07883-f001:**
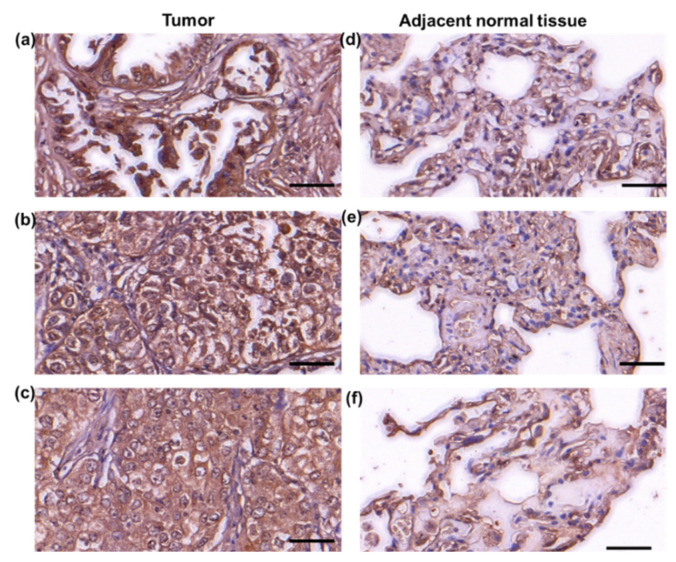
Increased RPIA expression was detected in the tumor biopsies of lung adenocarcinoma patients. Immunohistochemical staining was performed with anti-RPIA antibody on the lung adenocarcinoma tumor tissue arrays. (**a**–**c**) The adenocarcinoma grade 2 tumor tissues from 3 different lung adenocarcinoma patients, (**d**–**f**) the adjacent normal lung tissues from the same 3 patients in (**a**–**c**), respectively. Scale bar represents 50 μm.

**Figure 2 ijms-23-07883-f002:**
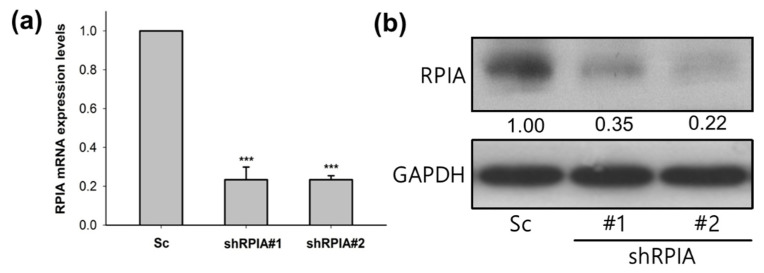
The mRNA and protein expression levels of RPIA were reduced in the two RPIA shRNA knockdown A549 lung cancer cells. (**a**) The RPIA mRNA levels were measured by Q-PCR in A549 lung cancer cells infected with either Scramble (Sc) control or shRNA (shRPIA#1 and shRPIA#2). Statistical analyses were done by Student *t*-test, *** *p* < 0.001. (**b**) The RPIA protein levels were examined by Western blot in A549 lung cancer cells infected with either Scramble (Sc) as control or shRNA (shRPIA#1 and shRPIA#2). GAPDH was used as an internal control.

**Figure 3 ijms-23-07883-f003:**
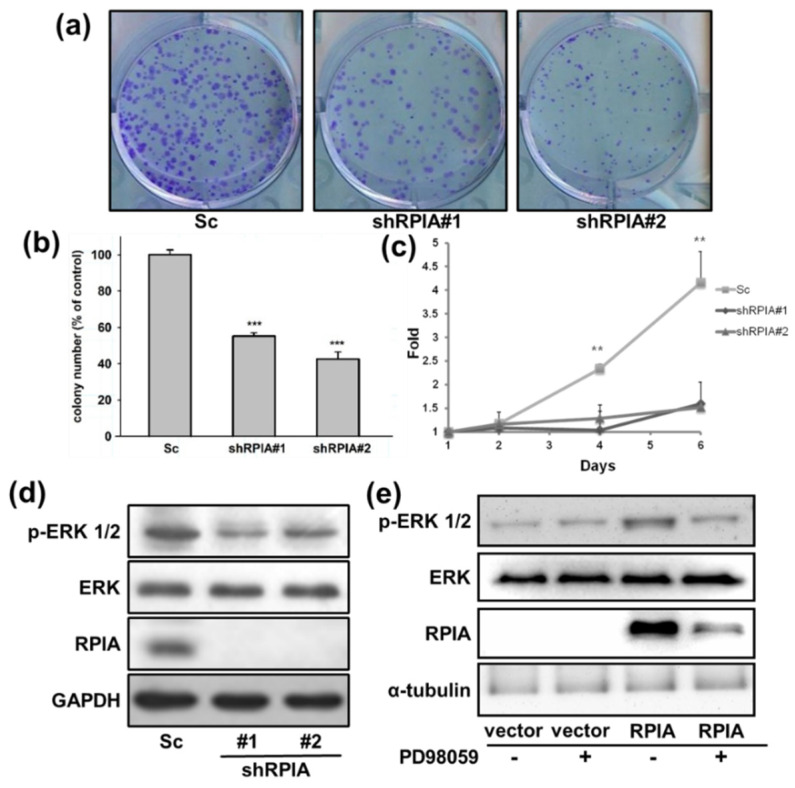
Knockdown of RPIA decreased the oncogenicity of A549 cells. (**a**) For colony formation assay, shRPIA-A549 cells were seeded at a density of 500 cells per well. After 10-day incubation, the cells were stained with crystal violet and the images were photographed for quantitation. (**b**) The colony numbers were further quantified and presented as the relative percentage compared to Scramble (Sc). Statistical analyses were conducted by Student *t*-test, *** *p* < 0.001. (**c**) Knockdown of RPIA decreased cell proliferation by WST-1 assay. Statistical analyses were done by Student *t*-test, ** *p* < 0.01. (**d**) Knockdown of RPIA decreased the levels of phosphorylated ERK1/2 (p-ERK1/2) in A549 lung cancer cell lines by Western blot. (**e**) The elevated p-ERK1/2 levels by overexpression of RPIA in A549 can be reduced by the treatment of PD98059 (10 μM), an MEK inhibitor.

**Figure 4 ijms-23-07883-f004:**
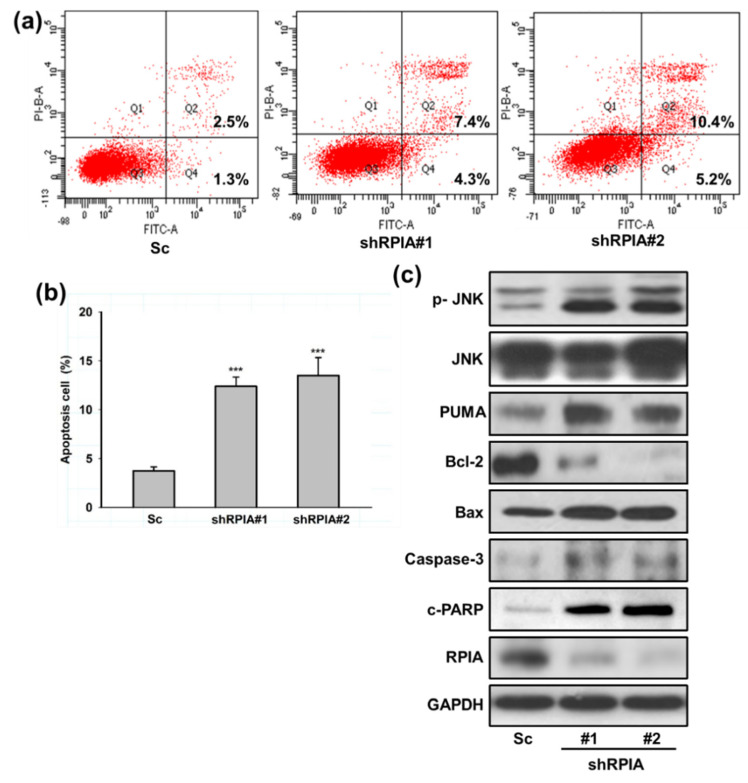
Knockdown of RPIA induced mitochondria mediated apoptosis in lung cancer cells. (**a**) Detection of apoptosis with the Annexin V-FITC/PI assay by flow cytometry. (**b**) Quantification of the ratio of the dead cells (PI positive, AnnexinV-FITC positive, Q2) versus the early apoptotic cells with intact membranes (PI negative, Annexin V-FITC positive, Q4), *** *p* < 0.001. (**c**) Knockdown of RPIA induced the expression of mitochondria mediated apoptosis related proteins (p-JNK, PUMA, Bcl-2, Bax, Caspase-3, cleaved-PARP) in the A549 cells by Western blot. GAPDH was used as an internal control.

**Figure 5 ijms-23-07883-f005:**
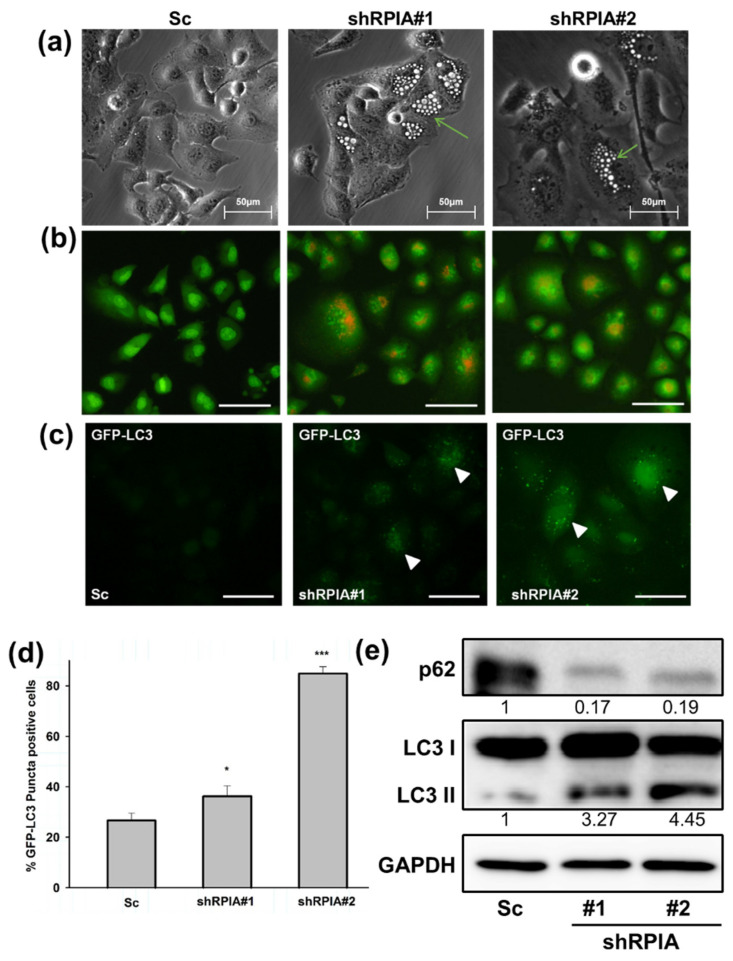
Knockdown of RPIA induced autophagosome formation, reduced p62 and increased LC3-II protein levels in lung cancer cells. (**a**) Knockdown of RPIA triggered autophagic vacuoles morphology observed by microscopy. Magnification: ×200. (**b**) Suppression of RPIA in A549 cells increased acridine orange staining detected by fluorescence microscopy. Magnification ×200, Scale bar represents 200 μm. (**c**) RPIA knockdown lung cancer cell displayed more LC3-GFP punctae by fluorescence microscopy. Magnification ×200. Scale bar represents 200 μm. (**d**) The percentage of the LC3-GFP puncta-positive cells was quantified and the statistical analyses were performed by Student *t*-test, * *p* < 0.05, *** *p* < 0.001. (**e**) Knockdown of RPIA increased LC3-II and reduced p62 expression in lung cancer cells.

**Figure 6 ijms-23-07883-f006:**
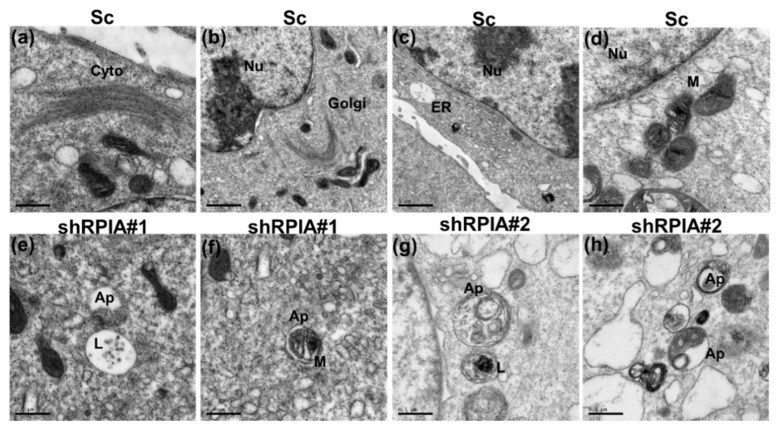
The appearance of autophagosome formation in shRPIA knockdown lung cancer cells by transmission electron microscopy. (**a**–**d**) Scramble control (sc), (**e**,**f**) A549 shRPIA#1, (**g**,**h**) A549 shRPIA#2. Nuclear (Nu), cytoskeleton (Cyto), mitochondria (M), Endoplasmic reticulum (ER), Golgi, autophagosome (Ap), lysosome (L). Magnification ×50,000.

**Figure 7 ijms-23-07883-f007:**
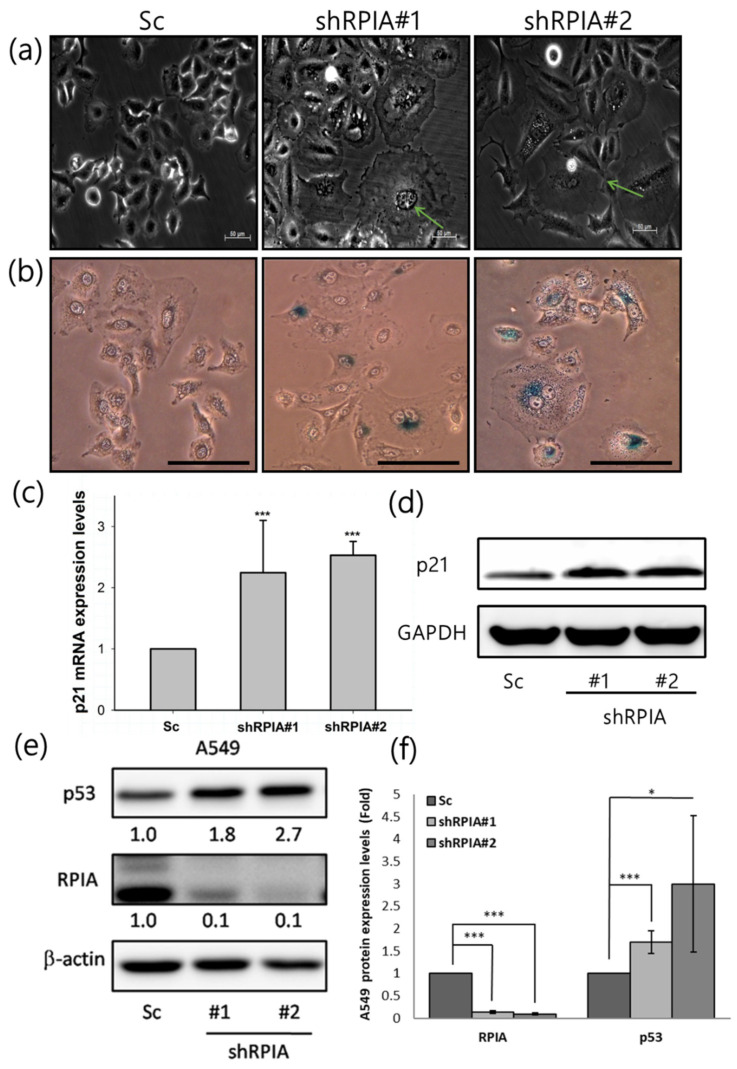
Knockdown of RPIA triggered cell senescence and increased the p53 and p21 protein levels in A549 cells. (**a**) Suppression of RPIA triggered cell senescence morphology detected by microscopy. The senescent cell was pointed out by the green arrow. Magnification: ×200. (**b**) Senescence-associated β-galactosidase activity (SA-β-gal) was enhanced in the RPIA knockdown A549 cells. Magnification: ×200, Scale bar represents 200μm. (**c**) The increased mRNA levels of p21 were detected by Q-PCR in the RPIA knockdown A549 cells. Statistical analyses were performed by Student *t*-test, *** *p* < 0.001. (**d**) The elevated protein levels of p21 were measured by Western blot in the RPIA knockdown A549 cells. GAPDH was used as an internal control. (**e**) The increased protein levels of p53 were detected in the RPIA knockdown A549 cells. β-actin was used as an internal control. (**f**) The folds of increase in p53 protein in the RPIA knockdown A549 cells. Statistical analyses were performed by Student *t*-test, *** *p* < 0.001, * *p* < 0.05.

**Figure 8 ijms-23-07883-f008:**
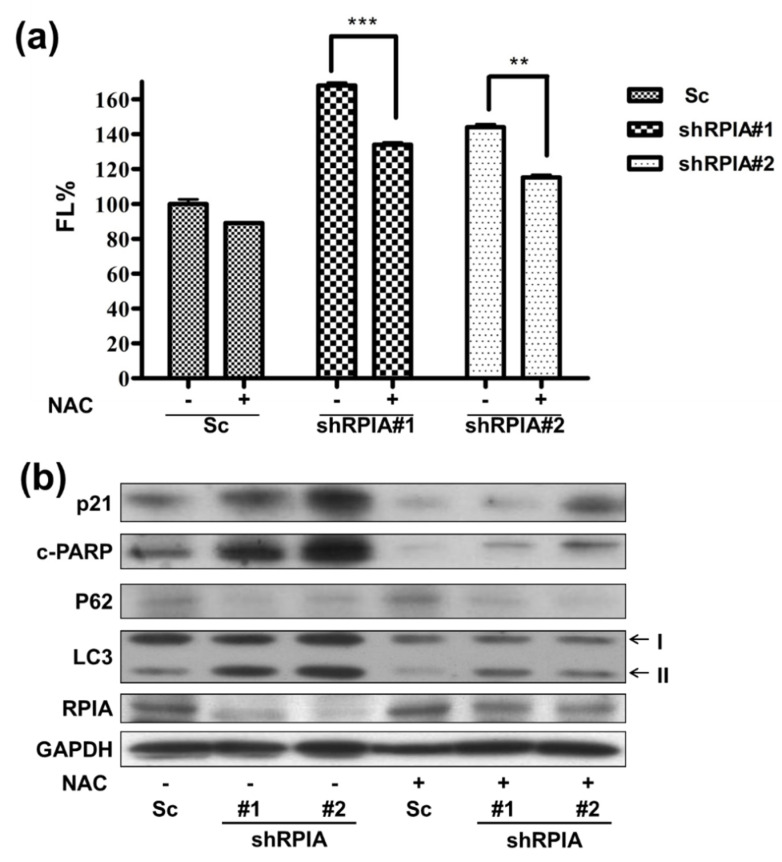
Reactive oxygen species (ROS) induced by RPIA knockdown lead to autophagy and apoptosis in lung cancer cells. (**a**) Treatment of ROS scavenger N-acetyl-L-cysteine (NAC) 10 mM in shRPIA-A549 cells were performed for one hour and the cells were stained with oxidized DCFDA, and then analyzed by flow cytometry. Knockdown of RPIA increased ROS generation. After treatment of NAC, shRPIA-A549 cells decreased ROS generation. FL% (DCF fluorescence). Statistical analyses were performed by Student *t*-test, ** *p* < 0.01, *** *p* < 0.001. (**b**) Knockdown of RPIA induced apoptosis, enhanced cleaved PARP (c-PARP), activated autophagy (up-regulated LC3-II, down-regulated p62) and cellular senescence (increased of p21) measured by using Western blot. After treating cells with ROS scavenger N-acetyl-L-cysteine (NAC) for 1 h, the knockdown of RPIA decreased autophagy and apoptosis protein marker expression in A549 cells.

**Figure 9 ijms-23-07883-f009:**
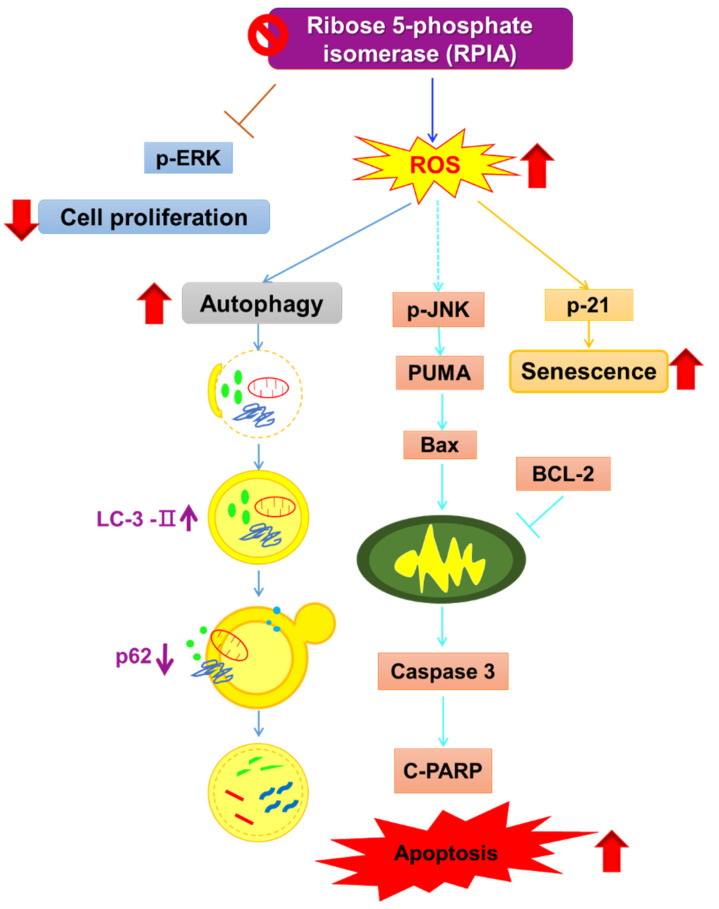
The working model for the molecular mechanisms of RPIA suppression activates apoptosis, autophagy and cellular senescence. The RPIA knockdown-induced autophagy is cytoprotective and a resistant mechanism against apoptosis.

**Table 1 ijms-23-07883-t001:** Q-PCR primer.

18 s	Forward	5′-GGA GAG GGA GCC TGA GAA AC -3′
	Reverse	5′-TCG GGA GTG GGT AAT TTG C -3′
RPIA	Forward	5′-GAT GCT GAT CTC AAT CTC ATC AA -3′
	Reverse	5′-GCA TAG CCA GCC ACA ATC TT-3′
p21	Forward	5′-TCACTGTCTTGTACCCTTGTGC-3′
	Reveres	5′-GGCGTTTGGAGTGGTAGAAA-3′

**Table 2 ijms-23-07883-t002:** Antibody list.

Antibody	Company	Cat-Number
RPIA	Abcam	ab67080
GAPDH	GeneTex	GTX100118
ERK1/2 MAPK	GeneTex	GTX59618
α-tubulin	GeneTex	GTX11312
Erk1(pT202/pY204)+ ERK2(pT185/pY187)	abcam	ab32538
LC3A/B	Cell Signaling	4108
SQSTM1	GeneTex	GTX100685
cleaved-PARP(c-PARP)	Cell Signaling	5625
Caspase-3	Epitomics	1476-S
Bax	Epitomics	1063-S
Bcl-2	Epitomics	1017-1
PUMA	GeneTex	GTX109675
p21	Santa Cruz	sc-397
phospho-JNK	Millipore	07-175
SAPK/JNK	Cell Signaling	9258

## Data Availability

Not applicable.

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
