# Peer review of "Suppression of Ribose-5-Phosphate Isomerase a Induces ROS to Activate Autophagy, Apoptosis, and Cellular Senescence in Lung Cancer"

_ijms, 2022, doi:10.3390/ijms23147883_

Round 1

Reviewer 1 Report

In this manuscript, the authors suggested that the suppression of RPIA induced autophagy, apoptosis, and cellular senescence via ROS generation in lung cancer cells. Although the authors performed various experiments to support their conclusion, this manuscript is not suitable for publication because of the following points.

-Major points-

1. The authors presented fragmentary results disconnected each other. How are the keywords (e.g. cell proliferation, autophagy, apoptosis, senescence) connected each other? The story is so confusing because the mechanistic study is insufficient.

2. What is the role of the RPIA knockdown-induced autophagy? Is it cytoprotective? Or does it lead to the autophagic cell death? It should be determined.

3. You cannot ascertain that knockdown of RPIA triggers cellular senescence because the evidence of senescence is too weak. The enhanced β-galactosidase staining is the sole evidence of senescence. P21 can be upregulated not only in senescence but also in transient cell cycle arrest. More evidence that supports your conclusion is needed. In addition, you did not determine the role of senescence.

4. In Figure 3e, how can RPIA be down-regulated by PD98059? This result suggest that the expression of RPIA can be regulated by ERK, which is opposite to your hypothesis that RPIA is an upstream regulator of ERK. In addition, the role of ERK in apoptosis, autophagy, and senescence should be investigated to complete the whole story.

-Minor points-

1. Introduction is too long. Please make the Introduction section more brief and concise.

2. The Figure legend in line 160-161 is confusing.

3. The subtitle in line 182 should be changed.

Author Response

We have revised our manuscript according to the reviewer's suggestions and comments and replied the questions point by point in the attached file.

We sincerely appreciate the reviewer's constructive comments and suggestions to substantially improve the quality of our revised manuscript. We hope our updated information and corrections will answer the questions from the reviewer. Thank you very much again for the assistance.

Reviewer 2 Report

Technical Quality: High

Novelty: High

Length: Suitable

Opinion about the manuscript: Is in need of significant revision

Opinion after Revision: Reconsider after major revision.

In this paper, the authors reported that inhibition of RPIA induces ROS in lung cancer and activates autophagy, apoptosis, and cellular senescence. ROS induction was confirmed by DCFA-DA. Apoptosis and autophagy were confirmed by in vitro and in vivo experiments using representative factors such as western blot, FACS, and TEM. However, the manuscript could be further strengthened with a few additional experiments denoted below.

1. In line 128, the relationship between autophagy and intercellular senescence in increasing JNK is insufficient.

2. In Figure 1, it would be better if the number 50 was clearly drawn on the scale bar.

3. In Figure 4A, the amount of the FACS control group was smaller than that of the shRPIA group. It does not seem to be the same experimental sample.

4. In Figure 4C, caspase 3 seems to be increased in the shRPIA group compared to the control group. Most of the data already showed that knockdown of RPIA induced apoptosis. Authors need to explain about that.

5. It would be good if the numbers are clearly visible on the scale bar in Figure 5B and C

6. There is a space error in Figure 7A on line 282.

7. In line 414, HT29 is a colon cancer cell, not a lung cancer cell. Make sure to check and correct it.

Author Response

We have revised our manuscript according to the reviewer's suggestions and comments and replied the questions point by point in the attached file.

We sincerely appreciate the reviewer's constructive comments and suggestions to substantially improve the quality of our revised manuscript. We hope our updated information and corrections will answer the questions from the reviewer. Thank you very much for the kind assistance.

Round 2

Reviewer 1 Report

Thank you for revising the manuscript as the reviewer’s comments. However, there are still some points that should be considered.

-Major points-

1. In Figure 9, the relationship between ERK and ROS is unclear even though the other keywords were related with ROS generation. Is ERK regulation independent on ROS? You can answer this question simply by detecting the expression of p-ERK in Figure 8B. If there is some relationship, please include that in Figure 9.

2. Please investigate whether NAC treatment reversed the RPIA knockdown-induced apoptosis.

-Minor points-

As regards the review round 1,

Reviewer’s comment 2. What is the role of the RPIA knockdown-induced autophagy? Is it cytoprotective? Or does it lead to the autophagic cell death? It should be determined.

Authors’ Reply: … The treatment of autophagy inhibitors by 3-methyladenine and chloroquine can further increase the levels of C-PARP compared to the untreated control (as shown below). We have added this data into the Figure S3 mentioned in Discussion in line 379.

à I think the description should be included in Results section instead of Discussion section. In addition, please revise Figure 9 to clearly mark the relationship between autophagy and apoptosis (cytoprotective and resistant mechanism against apoptosis).

Reviewer’s comment 3. You cannot ascertain that knockdown of RPIA triggers cellular senescence because the evidence of senescence is too weak. The enhanced β-galactosidase staining is the sole evidence of senescence. P21 can be upregulated not only in senescence but also in transient cell cycle arrest. More evidence that supports your conclusion is needed. In addition, you did not determine the role of senescence.

Reply: … We will be happy to add this data into Figure 7 to become Figure 7 e, f if the Editor and Reviewers consider these data should be included. …

à Please include the results in Figure 7e.

Author Response

We would like to thank Reviewer 1 for the constructive comments and suggestions. We have replied the questions point by point and added the data into the manuscript and edited as suggested accordingly.

Round 3

Reviewer 1 Report

The manuscript was fully revised according to the reviewer's consideration. I think the revised manuscript is now suitable for publication.